# Experimental Pathogenicity of H9N2 Avian Influenza Viruses Harboring a Tri-Basic Hemagglutinin Cleavage Site in Sonali and Broiler Chickens

**DOI:** 10.3390/v15020461

**Published:** 2023-02-07

**Authors:** Jahan Ara Begum, Ismail Hossain, Mohammed Nooruzzaman, Jacqueline King, Emdadul Haque Chowdhury, Timm C. Harder, Rokshana Parvin

**Affiliations:** 1Department of Pathology, Faculty of Veterinary Science, Bangladesh Agricultural University, Mymensingh 2202, Bangladesh; 2Institute of Diagnostic Virology, Friedrich-Loeffler-Institute, Federal Research Institute for Animal Health, Suedufer 10, 17493 Greifswald-Insel Riems, Germany

**Keywords:** LPAI, molecular characterization, pathogenicity, lesions, viral shedding, RT-qPCR

## Abstract

Low-pathogenic avian influenza (LPAI) H9N2 virus is endemic in Bangladesh, causing huge economic losses in the poultry industry. Although a considerable number of Bangladeshi LPAI H9N2 viruses have been molecularly characterized, there is inadequate information on the pathogenicity of H9N2 viruses in commercial poultry. In this study, circulating LPAI H9N2 viruses from recent field outbreaks were characterized, and their pathogenicity in commercial Sonali (crossbred) and broiler chickens was assessed. Phylogenetic analysis of currently circulating field viruses based on the hemagglutinin (HA) and neuraminidase (NA) gene sequences revealed continuous circulation of G1 lineages containing the tri-basic hemagglutinin cleavage site (HACS) motif (PAKSKR*GLF) at the HA protein. Both the LPAI susceptible Sonali and broiler chickens were infected with selected H9N2 isolates A/chicken/Bangladesh/2458-LT2/2020 or A/chicken/Bangladesh/2465-LT56/2021 using intranasal (100 µL) and intraocular (100 µL) routes with a dose of 10^6^ EID_50_/_mL_. Infected groups (LT_2-So1 and LT_56-So2; LT_2-Br1 and LT_56-Br2) revealed no mortality or clinical signs. However, at gross and histopathological investigation, the trachea, lungs, and intestine of the LT_2-So1 and LT_56-So2 groups displayed mild to moderate hemorrhages, congestion, and inflammation at different dpi. The LT 2-Br1 and LT 56-Br2 broiler groups showed nearly identical changes in the trachea, lungs, and intestine at various dpi, indicating no influence on pathogenicity in the two commercial bird species under study. Overall, the prominent lesions were observed up to 7 dpi and started to disappear at 10 dpi. The H9N2 viruses predominantly replicated in the respiratory tract, and higher titers of virus were shed through the oropharyngeal route than the cloacal route. Finally, this study demonstrated the continuous evolution of tri-basic HACS containing H9N2 viruses in Bangladesh with a low-pathogenic phenotype causing mild to moderate tracheitis, pneumonia, and enteritis in Sonali and commercial broiler chickens.

## 1. Introduction

Low-pathogenic avian influenza (LPAI) H9N2 virus has been endemic in Bangladesh since 2006 and causes significant economic losses in the poultry industry [1]. H9N2 virus infection induces mild respiratory illnesses such as coughing, sneezing, rales, and excessive lacrimation but can also run asymptomatic courses, especially in wild bird species [2,3]. Other clinical signs include a reduction in egg production (14–75%) in breeder or layer flocks with a variable mortality rate (10–60%) in chickens [4,5], which is accelerated by mixed infections [6,7].

Since the first outbreak in 2006, H9N2 viruses of the G1 lineage have been frequently identified in clinical samples from commercial and backyard poultry and in surveillance samples from live bird markets (LBMs) in Bangladesh [8,9,10]. Molecular characterization of the LPAI H9N2 viruses circulating in Bangladesh showed accumulating mutations that favor interspecies transmission [11]. In avian hosts, the pathogenicity of avian influenza virus (AIV) subtypes is strongly correlated with the HA endoproteolytic cleavage site (HACS). The H9N2 viruses in Bangladesh initially displayed a di-basic (PAKSSR*GLF) cleavage site, typical of viruses of the G1 lineage of H9N2 [11]. However, later strains with a tri-basic (PAKSKR*GLF) HA cleavage site motif are being identified with increasing frequency [12]. 

In recent years, H9N2 virus has attained further importance due to its involvement as a donor in the reassortment of zoonotic AIV of subtypes H5, H7, and H10 [13]. H9N2 viruses of the G1 lineage also have intrinsic zoonotic potential [5]. Several in vivo and in vitro studies have been conducted to understand the pathogenesis of H9N2 in chickens [2,3]. In some experimental studies, H9N2 infections did not produce obvious clinical signs or death, and replication of the virus was limited to the upper respiratory tract and digestive tract in specific pathogen-free (SPF) chickens under laboratory conditions [2,3]. In contrast, some studies reported that H9N2 causes moderate to severe respiratory infections and low mortality in commercial broiler chickens with replication of the virus in multiple tissues [3]. An in ovo pathogenesis study revealed that tri-basic H9N2 viruses replicated in a grossly extended spectrum of embryonic organs [14]. These discrepancies may be explained by pathogenic diversity across H9N2 strains, including molecular determinants linked to AIV pathogenicity in chickens [2,3], or by secondary infections in the field. 

In Bangladesh, commercial poultry farming has been growing rapidly since the early 1990s [15]. Poultry farming, including broilers and layers, has created employment, improved food security, and enhanced the supply of quality protein to people’s meals, thus contributing to the country’s economic growth [15,16]. Recently, the Sonali chicken, a local crossbreed of Rhode Island Red (RIR) cocks and Fayoumi hens, has gained popularity for meat purposes among the farmers as it is well adapted to the environmental conditions of Bangladesh and requires less care and attention when compared with other breeds. However, the frequent outbreaks of LPAI H9N2 in association with other avian respiratory infections cause huge economic losses, and mitigation is sometimes complicated by incorrect diagnosis due to the lack of typical clinical symptoms in infected flocks [17]. Although pathological changes in H9N2 AIV-affected broilers and layers have been extensively investigated, worldwide, very little is known about the progressive development of pathological changes in H9N2 AIV-infected local chicken breeds such as Sonali. Therefore, in this study, recently circulating H9N2 AIVs of lineage G1 and expressing a tri-basic HACS were used to experimentally infect Sonali and commercial broilers (Cobb 500) for comparison. Birds were monitored for clinical outcomes, gross and microscopic changes in tissues, and virus shedding at different days post infection (dpi) under experimental conditions.

## 2. Materials and Methods

### 2.1. Sample Collection and Virus Detection by Real Time RT-PCR (RT-qPCR)

Trachea and lung tissue samples (LT_2, LT_14, LT_56, and LT_86) obtained at necropsy of affected chickens were collected from four commercial poultry farms in Mymensingh division in the North of Bangladesh during 2020 and 2021. All sampled flocks had a history of respiratory distress, and hemorrhage, and congestion in the trachea, lung, and pancreas (Appendix A). Tissue samples were collected in sterile tubes, transported in a cooling device, and stored at −80 °C until further use. 

Viral RNA was extracted using the QIAamp Viral RNA Mini Kit (QIAGEN, Hilden, Germany) following the manufacturer’s instructions and quantified using a Nanodrop One (Thermo Scientific, Waltham, MA, USA). RNA was then tested for the Matrix (M) gene of AIV and for HA H9 and NA N2 subtype-specific segments by an established RT-qPCR utilizing a set of gene-specific primers [18,19]. Luna^®^ universal probe one-step RT-PCR kit (Biolabs^®^, New England) was used to prepare 12.5 µL volume of reaction mixture containing the following: 6 µL master mix, 1 µL forward primer (10 pmol/µL), 1 µL reverse primer (10 pmol/µL), 0.5 µL enzyme mix, 1.5 µL RNase-free water, and 2.5 µL of extracted RNA. The reaction was conducted in a 7500 fast real-time PCR machine (Thermo Scientific, Waltham, MA, USA) using the following thermal profile: reverse transcription at 55 °C for 2 min, initial denaturation at 95 °C for 10 min, followed by 40 cycles of denaturation (95 °C, 15 s) and annealing (60 °C, 1 min).

### 2.2. Virus Isolation

Selected RT-qPCR-positive H9N2 samples were propagated in the allantoic cavity of 10-day-old embryonated chicken eggs. The eggs were incubated at 37 °C for 96 h. Subsequently, allantoic fluid (AF) was harvested and stored at −80 °C until use. The harvested AF was subjected to a hemagglutination test (HA) using 1% chicken red blood cells (RBCs) using standard procedures [20]. The presence of the virus in HA-positive AFs was confirmed by RT-qPCR. 

### 2.3. Sequencing and Phylogenetic Analysis

A nanopore-based amplification method was utilized for full-genome sequencing of AIV H9N2-positive allantoic fluids as previously described [21]. Briefly, universal whole-genome amplification of the extracted RNA was conducted by AIV-End-RT-PCR with the Superscript III One-Step RT-PCR System (ThermoFisher Scientific, Rochester, NY, USA). Afterwards, the PCR products were purified with AMPure XP Magnetic Beads (Beckman-Coulter, Brea, CA, USA). Full-genome sequencing on a Mk1C MinION platform (Oxford Nanopore Technologies, ONT, Oxford, UK) in combination with a R9.4.1 flow cell and the Rapid Barcoding Kit (SQK-RBK004, ONT) was conducted according to the manufacturer’s instructions. MiniMap2 [22] is employed in full-genome consensus sequence production in a map-to-reference approach. The final genome sequences were polished manually after consensus production according to the highest quality (60%) in Geneious Prime (Biomatters, New Zealand). Contemporary Bangladeshi H9N2 along with other representative field strains were downloaded from a public domain (NCBI GenBank or the Global Initiative on Sharing All Influenza Data (GISAID) database). The selected sequences were then subjected to multiple alignment using MAFFT software online version (https://mafft.cbrc.jp/alignment/server/; accessed on 13 September 2022). Phylogenetic analysis was performed using the neighbor joining method and Jukes-Cantor substitution model incorporated in the MAFFT package, with the confidence intervals estimated by applying a 1000 bootstrap algorithm [23]. Finally, phylogenetic trees were annotated and visualized using FigTree v1.4.2 software (http://tree.bio.ed.ac.uk/software/figtree/; accessed on 14 September 2022) and Inkscape 1.0 (https://inkscape.org; accessed on 14 September 2022). The obtained full genome sequences of four Bangladeshi H9N2, A/chicken/Bangladesh/2458-LT2/2020 (LT_2), A/chicken/Bangladesh/2462-LT14/2020 (LT_14), A/chicken/Bangladesh/2465-LT56/2021 (LT_56), and A/chicken/Bangladesh/2474-LT86/2021 (LT_86) were deposited in the GISAID database (Appendix A). The molecular genetic properties or amino acid mutation profiles of HA proteins were compared with the selected Bangladeshi strains and the reference G1 strain (EPI985025_A/Quail/HK/G1/1997) using the web-based tool FluSurver incorporated into the GISAID platform.

### 2.4. Experimental Pathogenesis Study of H9N2 AIVs

#### 2.4.1. Virus Titration and EID_50_ Calculation

Among the respective four H9N2 field isolates, LT_2 and LT_56 were selected for the experimental pathogenesis study as they were obtained in 2020 and 2021. The 50% egg infectious dose (EID_50_) of the virus stocks was calculated based on serial 10-fold dilutions in PBS and culture in 10-day-old embryonated chicken eggs following standard procedures [24].

#### 2.4.2. Experimental Design

For the animal experiment, day-old chicks of Sonali (*n* = 70) and broiler (*n* = 70) breeds were obtained from a leading, trusted commercial company and raised with food and water ad libitum. At the age of 42 (Sonali) and 21 (broiler) days, AIV-seronegative chickens were divided into three groups for each bird type: the LT_2 infected Sonali group (LT_2-So1) (*n* = 30), LT_56 infected Sonali group (LT_56-So2) (*n* = 30), and control Sonali group (So3) (*n* = 10); similarly, broilers were distributed to LT_2 infected broiler (LT_2-Br1) (*n* = 30); LT_56 infected broiler (LT_56-Br2) (*n* = 30) and a control broiler group (Br3) (*n* = 10) (Figure 1). The different groups were housed separately. The chickens of each infected group (LT_2-So1, LT_56-So2, LT_2-Br1 and LT_56-Br2) were inoculated with 200 µL of 10^6^ EID_50_/_mL_ [25,26] of the LT_2 and LT_56 isolates via intranasal (100 µL) and intraocular (100 µL) routes at 42 days of age for Sonali and at 21 days of age for broiler. Control groups (So3 and Br3) received 200 µL of phosphate buffer saline (PBS) via the same routes. Birds were monitored daily for clinical signs until 15 days post-inoculation (dpi). Each of the five birds per group were euthanized at 1, 3, 5, 7, 10, and 15 dpi. Before euthanasia, oropharyngeal and cloacal swabs were collected in 1 mL PBS containing antibiotics (Gentamycin @ 500 µg/mL) and stored at −80 °C until subsequent analysis. Blood samples were collected from the infected and control birds at 0, 7, and 14 dpi. 

#### 2.4.3. Clinical and Pathological Investigation

The birds in each group were inspected twice a day for any behavioral changes or clinical manifestations. A full necropsy was performed on each bird. At necropsy, tissue samples (trachea, lungs, and intestine) were collected in 10% neutral buffered formalin for microscopic examination and in sterile tubes for virus detection. Subsequently, the formalin-fixed tissues were embedded in paraffin. The tissue sections were prepared and stained with hematoxylin and eosin (H&E) stain as per the standard method [27]. The stained slides were examined using a light microscope, an Olympus CX43 equipped with an EP50 camera (Olympus Corporation, Shinjuku City, Japan).

#### 2.4.4. Viral Distribution and Shedding

To assess the viral distribution and shedding in the trachea, lungs, and intestine of necropsied birds as well as in oropharyngeal and cloacal swabs of dpi 1, 3, 5, 7, 10, and 15, tissue suspensions or swabs were prepared in 2 mL of minimal essential medium supplemented with penicillin and streptomycin. A single stainless-steel bead (5 mm) was added to each organ sample and homogenized in a 2 mL collection tube for 2 min using a Tissue Lyser instrument (Qiagen, Hilden, Germany). RNA was extracted from both tissue suspensions and swab samples. RT-qPCR was performed as described above. Cycle threshold (Ct) value was used to semi-quantify the viral load after RT-qPCR targeting the H9 gene. A Ct value of <40 was used as a threshold for positive results. 

#### 2.4.5. Serology

Hemagglutination inhibition (HI) assays were performed following standard guidelines [20]. Sera were obtained by centrifuging blood samples at 3000 rpm for 10 min, inactivated at 56 °C for 30 min, and stored at −20 °C until used. Sera were titrated for H9-specific antibodies by HI assay against four hemagglutination units (4HA) of the respective homologous H9N2 antigens.

#### 2.4.6. Statistical Analyses

Statistically significant differences between Ct values from the H9-specific RT-qPCR obtained at different organs of experimentally infected LT_2 and LT_56 in Sonali and broiler chickens were determined by ANOVA. A one-way ANOVA with Tukey’s multiple comparison test was performed to elicit whether the antibody titers varied significantly among the pre- and post-infected groups. Results were plotted using GraphPad Prism 5 (La Jolla, CA, USA).

## 3. Results

### 3.1. Genetic Background of Recently Circulating H9N2 Viruses in Bangladesh

Four strains (LT_2, LT_14, LT_56, and LT_86) of low-pathogenic H9N2 viruses were successfully propagated in embryonated chicken eggs, identified by RT-qPCR, and fully sequenced. All eight genome segments were characterized. Phylogenetic analysis based on the HA and NA gene sequences revealed characteristics of the G1 lineage of H9N2 viruses in Bangladesh (Figure 2). The studied isolates were closely (>98.9%) related to other contemporary H9N2 viruses from Bangladesh and contained the tri-basic HACS motif (PAKSKR*GLF). The six internal genes also exhibited close similarities to current Bangladeshi strains; however, multiple clustering in their phylogenetic trees was evident (Appendix A). The HA protein of the recent Bangladeshi strains showed amino acid (aa) mutations characterizing the antigenic properties of the strains. Unique mutations of 153N in isolates LT 2 and LT 86, 198T in LT 14, and 167S in all four isolates were found as potential antigenic sites (Table 1). Other antigenic sites showed previously identified mutations, while some sites remained conserved. Two additional aa mutations, V287A and I306V, were found in the LT 56 HA gene and were absent in LT 2, LT 14, and LT 86, and are functionally related to binding small ligand(s), viral oligomerization interfaces, and antibody recognition sites. Furthermore, the four isolates studied had sporadic mutations in their HA that were not functionally related. At the NA protein, the studied isolates and other contemporary strains revealed genotyping differences as well as several unspecific mutations. Appendix A shows the detailed functions of the aa mutations found in the HA and NA proteins. 

### 3.2. Experimental Pathogenicity Study

#### 3.2.1. Clinical Signs, Gross and Histopathological Lesions

No mortality or clinical signs were observed in both infected and control Sonali or broiler chickens throughout the experimental periods. One control and five infected birds were euthanized at 1, 3, 5, 7, 10, and 15 dpi and examined at necropsy. At necropsy, the tracheas of infected Sonali chickens (LT_2-So1 and LT_56-So2) showed slight hemorrhages along the tracheal rings up to 15 dpi (Appendix A). In the lungs, there was mild to moderate progressive congestion and hemorrhages and mild focal fibrin deposition up to 7 dpi (Appendix A) in both H9N2 virus-infected groups of Sonali chickens. At 10 dpi, the lungs of the infected Sonali chickens began to recover. Similarly, the intestinal mucosa of the infected Sonali chickens showed mild to moderate progressive petechial hemorrhages up to 7 dpi, whereas at 10 and 15 dpi, intestine samples appeared unaltered (Appendix A). Similar gross pathological changes in the trachea and lungs were observed in the H9N2 infected broiler groups (LT_2- Br1 and LT_56-Br2) at the respective dpi. However, the intestinal hemorrhages were absent in infected broiler birds in comparison to the Sonali chickens.

Histopathological findings in the tracheas of infected Sonali chickens (LT_2-So1 and LT_56-So2) revealed slight proliferation of goblet cells from the beginning of the experiment, focal loss of lining epithelium and cilia, and slight infiltration of inflammatory cells at 3 dpi (Figure 3b,c). In Sonali chickens, tracheitis progressed with loss of mucosal epithelium from 5 dpi and continued until 10 dpi (Figure 3d,f). However, in broilers, tracheitis started at 3 dpi, continued until 5 dpi, and resolved thereafter. In Sonali chickens, mild pneumonia was evident at 1 dpi (Figure 4b). Severe hemorrhages and the collapsing of the alveoli with edema around blood vessels were found at 3 and 5 dpi (Figure 4c,d). At the initial stage, pneumocytes-I were lost and were gradually replaced by pneumocytes-II (Table 2). The lung tissues started resolving at 7 dpi and proliferation of pneumocytes-II was seen up to 15 dpi (Figure 4e,f; Table 2). In broiler chickens, moderate pneumonia developed at 5 dpi, which gradually became milder with time. At the same time, the loss of pneumocytes-I and proliferation of pneumocytes-II were evident (Table 2).

In the intestine of Sonali chickens, no histological lesions were observed on 1 dpi; however, at 3 dpi, intestinal villi were severely lost and crypts were visible. The desquamated epithelial cells were accumulated in the lumen of the intestine (Figure 5b). At 5 dpi, the crypt epithelia were found hyperplasic with huge goblet cell proliferation; desquamated epithelial cells were accumulated in the intestine (Figure 5c). A few short-length villi and crypts were observed at 5 dpi in the intestinal lumen, which may indicate the beginning of villi regeneration. The villi were found to be sufficient in length and height, healthy, and completely covered by tall columnar epithelial and goblet cells at 7 dpi (Figure 5d). The same lesions were found at 10 and 15 dpi. In broilers, similar lesions were observed.

Altogether, both infected groups of broiler chickens (LT_2-Br1 and LT_56-Br2) showed similar histopathological changes in the trachea and lungs as the Sonali chickens. The detailed histopathological scoring of different parameters of the trachea, lungs, and intestine of Sonali and broiler chickens infected with H9N2 is shown in Table 2.

#### 3.2.2. Respiratory and Intestinal Distribution and Shedding of the H9N2 Viruses

The RNA load of H9N2 viruses in the trachea, lungs, and intestine and their shedding via the oropharyngeal and cloacal routes of all infected groups (LT_2-So1, LT_56-So2, LT_2-Br1, and LT_56-Br2) were studied by RT-qPCR targeting the H9 gene segment. In LT_2 and LT_56 infected Sonali chickens, viral RNA was detected in the trachea, lungs, and intestine from 1 dpi to 10 dpi. The highest viral load was detected at 3 and 5 dpi in the trachea, lung, and intestine. The viral RNA load decreased progressively from 7 dpi, and no RNA was detected after 15 dpi. Similar viral dissemination kinetics were observed in the tissues of infected broiler chickens. The overall viral load in the lungs and intestine of Sonali chickens was incomparable to broiler chickens (Figure 6). However, the distribution of H9N2 viruses in the trachea is wider, and viral loads shed through the oropharyngeal route are significantly higher (*p* < 0.05) in the cases of LT_2-So1 and LT_56-So2 compared with the LT_2-Br1 group of birds, according to a two-way ANOVA with Bonferroni post hoc testing. The infected chickens shed viruses via oropharyngeal and cloacal routes. Viruses were detected in the oropharyngeal swabs between 1 and 10 dpi and in cloacal swabs between 3 and 10 dpi (Figure 6). No significant differences were observed in terms of viral RNA load and kinetics of dissemination between the two studied virus isolates (LT_2 and LT_56) in both infected chicken (Sonali and broiler) groups. 

#### 3.2.3. Seroconversion

All chickens were considered susceptible to AIV before infection. The kinetics of antibody formation by means of the hemagglutination inhibition (HI) test are presented in Appendix A. SPF chicks are not available in Bangladesh. Therefore, commercial day-old chicks had to be purchased. These showed a high level of maternally derived antibodies (MDA) as they hatched from vaccinated hens. The chicks were raised in isolation with strict biosecurity measures. Before challenge, AIV-specific antibody levels were measured again, and most of the birds were found seronegative. Very few chicks still showed detectable antibody titers, but they were well below the level assumed to be protective (>5 HI units). On the other hand, immediately following challenges, the protective level increased significantly at 7 and 14 dpi (Appendix A).

## 4. Discussion

Since 2006, the low-pathogenic avian influenza (LPAI) H9N2 virus has been endemic in Bangladesh, causing huge economic losses in the poultry industry. The current study describes the genetic characteristics of recently circulating H9N2 avian influenza viruses and their pathogenicity in crossbred Sonali and Cobb 500 broiler chickens under experimental conditions. Genetic analysis of H9N2 LPAIV has identified several distinct lineages, with the G1 lineage being the most prevalent and diversified. The G1 lineage has spread to different countries in Asia, Africa, and the Middle East [2]. H9N2 AIV isolated from recent poultry samples in Bangladesh clustered with the G1 lineage, and phylogenetic analysis of the study isolates revealed several clusters and branching with contemporary Bangladeshi H9N2 strains. The studied Bangladeshi H9N2 isolates from two consecutive years (2020 and 2021) harbored the tri-basic (PAKSKR*GLF) HACS that has evolved in H9N2 AIV since 2011 [12], whereas until 2010, all reported HA sequences encoded a mono- or dibasic cleavage site motif with K/RSSR*GLF [11,12]. In avian hosts, the HACS is one of the important factors contributing to viral pathogenicity. Coding mutations in HA and NA proteins of the studied H9N2 viral genome confirmed continuing evolution within the G1 lineage. In general, the amino acid mutations at HA and NA proteins are functionally involved in antigenic shift or drift, virulence, binding ligands, viral oligomerization interfaces, escape mutants, creating or removing a potential N-glycosylation site, drug binding or resistance, and other biological properties of the circulating strains [28,29]. However, a significant number of mutations in the currently studied isolates were found in the potential antigenic sites of the HA protein that may alter its antigenic properties and contribute to vaccinal escape (Table 1; Appendix A). This might also explain why H9N2 outbreaks, despite vaccination, have not decreased but rather continued to spread across the country.

Results of previous in vitro studies were unequivocal as to whether the evolution of a tri-basic HACS already induced phenotypic changes in vivo or whether a true furin-sensitive HACS motif such as R-X-K/R-R was required for trypsin-independent, systemic replication [14]. Here we took the opportunity to descriptively analyze the pathogenicity of currently circulating LPAI H9N2 isolates with a tri-basic HACS in Sonali and conventional broiler chickens. No obvious clinical signs and mortality were noted in H9N2 infected and control birds, supporting the findings observed by Subtain et al. [30]. However, H9N2 viruses of the G1 lineage expressing mono-basic HACS have caused up to 30% mortality in layers in other studies, and some H9N2-infected broilers have shown diarrhea and a slight depression without any respiratory clinical signs [30,31]. In general, respiratory manifestations were common in H9N2 infection under experimental conditions, which may exacerbate in association with other respiratory co-infections such as Newcastle disease, infectious bronchitis, colibacillosis, and mycoplasmosis [32,33]. 

Although no clinical signs were observed in any group of infected chickens, moderately severe pathological lesions were evident specifically in their respiratory systems (trachea and lungs), similar to previous studies [33,34], and less so in the gastrointestinal tracts. This correlated with virus excretion via the oropharynx and the cloaca, respectively. Enteritis with mild hemorrhages, desquamation of villi epithelium, and glandular proliferation of the crypt epithelium were observed histologically in the infected Sonali chickens only. According to our study findings, the target organs for experimental H9N2 virus infection in Sonali and broiler chickens were the trachea and lungs, causing tracheitis and pneumonia. Slemons et al. [35,36] did not report the involvement of the lungs when the infection was given through the intravenous route. Furthermore, several studies reported that H9N2 infected chickens produced prominent lesions in brain and kidneys in addition to respiratory and gastrointestinal tracts [3,30,33], which were absent in our study. 

Replication of LPAIVs is limited to trypsin-expressing epithelial cells lining the respiratory and gastrointestinal tracts [37,38]. In this study, we found that LT_2 and LT_56 H9N2 viruses of the G1 lineage replicated in the trachea, lungs, and intestine of Sonali and broiler chickens between 1 and 10 dpi. H9N2 viruses belonging to the Y280 lineage were reported to replicate in brain tissues following intravenous infection without causing clinical symptoms [3,39]; however, the current study lacks evidence for viral replication in other systemic organs (brain, spleen, liver, and thymus). Although there were no signs of disease in either Sonali or broiler chickens throughout the experimental period, there was a difference in the level of viral replication between the two bird types. Overall, viral loads in the trachea, lungs, and intestine of Sonali chickens were significantly higher in comparison to the broiler chickens. For both viruses, the highest viral load was seen at 3, 5, and 7 dpi, which gradually started to decline from 10 dpi, and eventually disappeared after 15 dpi. All the infected chickens shed higher virus loads via the oropharyngeal route than the cloacal route until 10 dpi. The lack of viral RNA found in cloacal swabs taken at 1 dpi indicated that there was no cloacal shedding on that particular dpi, and this is further supported by the histopathogical findings of the intestine, which show almost normal intestinal morphology. Furthermore, in this study, no significant differences were observed between the two studied isolates (LT_2 and LT_56) in either of the infected chicken groups. Broiler chickens overall seemed to be less vulnerable to H9N2 infection than Sonali chickens. Considering that H9N2 AIVs belonging to the G1 lineage have been circulating in commercial poultry, including broilers, since 2006, it is possible that they have evolved continuously and are better adapted to commercial poultry in Bangladesh.

## 5. Conclusions

The avian influenza H9N2 viruses prevailing in commercial poultry in Bangladesh belong to the G1 lineage and express a tri-basic HACS and multiple mutations at antigenic sites. The pathogenicity of two LPAI H9N2 subtype viruses isolated recently in Bangladesh was assessed in Sonali (crossbreed) and Cobb 500 broiler chickens. Under experimental conditions, both virus isolates did not produce any mortality or clinical signs in either of the infected bird types. The viruses demonstrated tropism mainly for the respiratory and gastrointestinal tracts, while more virus was shed through the oropharyngeal route up to 10 dpi. Virus shedding was found to be more prolonged in Sonali chickens. The presence of a tri-basic HACS, although only one mutation step away from a bona fide highly pathogenic HACS, did not increase the pathogenicity of the viruses.

## Figures and Tables

**Figure 1 viruses-15-00461-f001:**
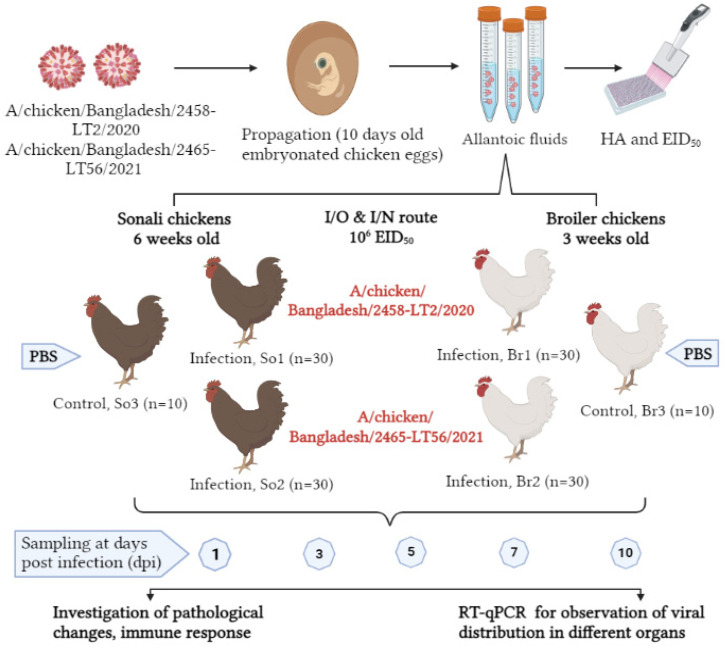
In vivo experimental design of H9N2 infection in Sonali and Cobb 500 broiler chickens using A/chicken/Bangladesh/2458-LT2/2020 (LT_2) and A/chicken/Bangladesh/2465-LT56/2021 (LT_56) isolates of Bangladeshi poultry.

**Figure 2 viruses-15-00461-f002:**
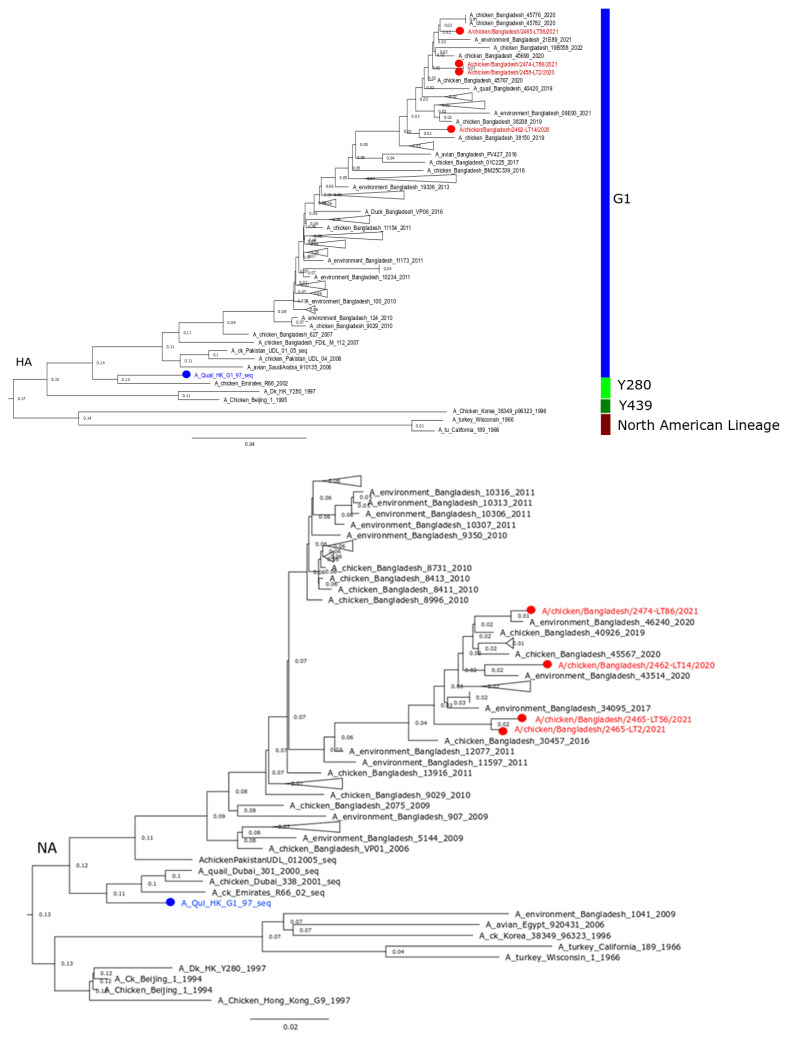
Phylogenetic tree for HA (**upper**) and NA (**lower**) genes of recent H9N2 isolates of Bangladesh. The tree was built using neighbor joining method and Jukes-Cantor substitution model incorporated in the MAFFT package with the confidence intervals estimated by applying 1000 bootstrap algorithm. Red color taxa with circle indicate studied Bangladeshi isolates. Blue color taxa represent G1 reference strain.

**Figure 3 viruses-15-00461-f003:**
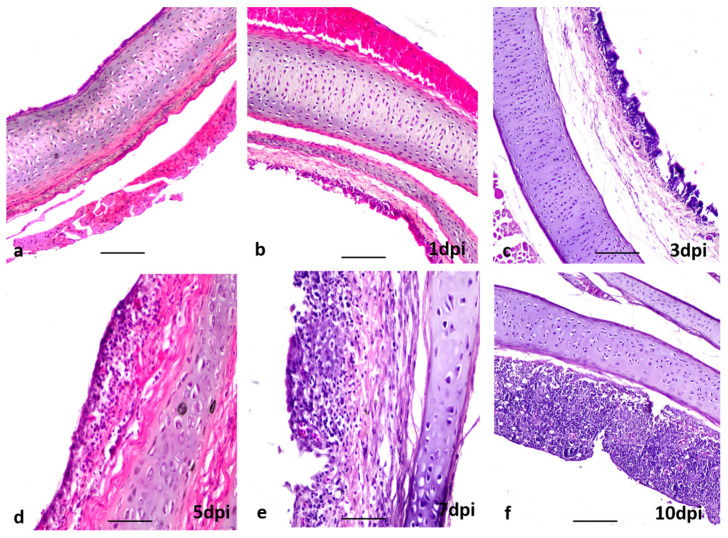
Section of trachea of chickens; (**a**) uninfected, no lesions, (**b**–**f**) infected with LPAI, (**b**), no lesions at 1 dpi, (**c**) mild tracheitis with partly sloughing of mucosa, (**d**) moderate to severe tracheitis, (**e**,**f**) severe tracheitis. H & E stain, bar: (**a**–**c**,**f**) =100 µm; (**d**,**e**) = 20 µm.

**Figure 4 viruses-15-00461-f004:**
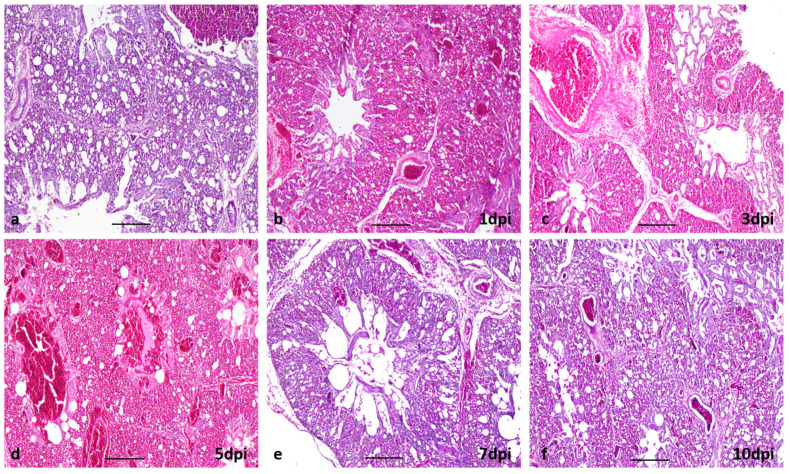
Section of lung of chickens (**a**) uninfected, no lesions, (**b**–**f**) infected with LPAI, (**b**), mild congestion, (**c**) congestion with edema, (**d**) severe congestion, edema and collapsing of alveoli, (**e**,**f**) almost normal appearance of lung, comparable to control (**a**). H & E stain, bar: (**a**–**d**,**f**) = 50 µm; (**e**) = 20 µm.

**Figure 5 viruses-15-00461-f005:**
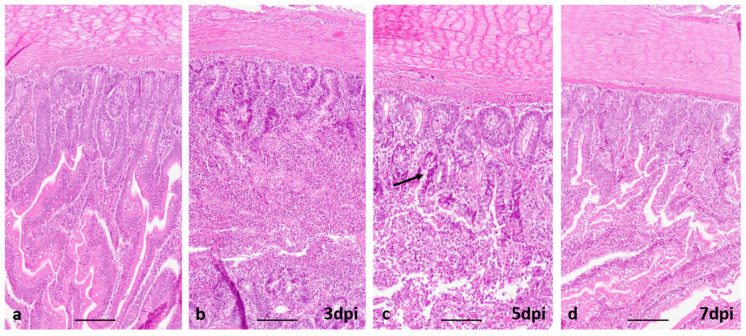
Section of small intestine of chickens. (**a**) uninfected, no lesions, (**b**) Severely damaged mucosa with huge desquamation and accumulation of lining epithelial cells in the intestinal lumen. (**c**) Hyperplasia of crypts epithelium (black arrow) and regeneration of villi. (**d**) The villi were found to be sufficient in length and height. H & E stain, bar: (**a**–**d**) = 100 µm.

**Figure 6 viruses-15-00461-f006:**
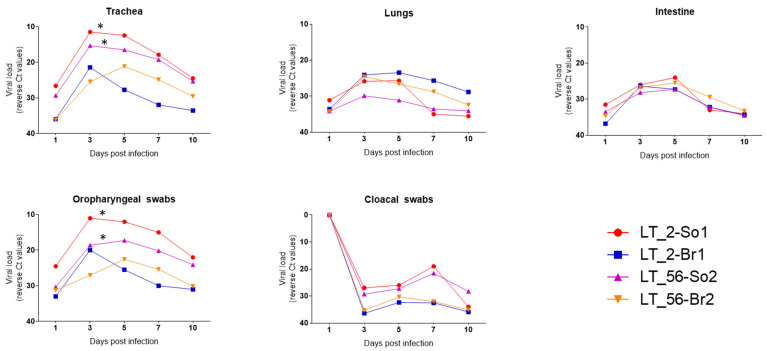
Kinetics of H9N2 virus replication and shedding upon experimental infections in Sonali and broiler chickens. The samples that showed value of >40 considered as negative and in the graph represent as zero (0). The graphs showed the Ct values generated at RT-qPCR targeting H9 gene in organs (trachea, lungs, intestine) and swabs (oropharyngeal and cloacal) sample. In general, virus replication progressively increased at 3 and 5 dpi whereas, progressively started to decline from 7 dpi up to 10 dpi. In most cases, viral load is higher in Sonali chicken compared with broiler. No significant differences between two viruses (LT_2 and LT_56) in each infected chicken group. Asterisk (*) indicates level of significance (*p* < 0.05).

**Table 1 viruses-15-00461-t001:** Differences in amino acid sequences in the HA protein of four Bangladeshi isolates, as well as other selected Bangladeshi strains and the G1 reference strain.

Sequence Accession & ID	Potential Antigenic Sites													RBS			HACS
H9 Numbering	66	90	147	149	153	167	168	196	198	216	224	282	283	166	234	399	335–338
EPI2187437_LT_86/2021	H	G	T	K	N	S	L	D	A	D	L	N	S	N	L	K	KSKR
EPI2187429_LT_56/2021	R	.	.	.	D	.	.	.	.	.	.	.	.	.	.	.	KSKR
EPI2187421_LT_14/2020	.	.	.	.	D	.	.	.	T	.	.	.	.	.	.	.	KSKR
EPI2187413_LT_2/2020	.	.	.	.	.	.	.	.	.	.	.	.	.	.	.	.	KSKR
EPI1778204_40818/2019	P	.	.	.	D	G	Q	.	.	N	.	.	.	.	.	.	KSKR
EPI1581761_35417/2018	R	.	.	.	D	G	.	.	.	N	.	K	.	.	.	.	KSKR
EPI1777263_NRL3238/2017	R	.	.	.	D	G	.	.	.	N	.	K	.	.	.	.	KSKR
EPI1508528_AR11758/2016	Q	.	.	.	D	G	Q	.	.	N	.	.	.	.	.	.	KSKR
EPI965466_24249/2015	R	.	.	.	D	G	.	.	.	N	.	.	.	.	.	.	KSKR
EPI963330_23727/2014	.	E	.	.	D	S	Q	.	.	N	.	R	.	.	.	.	KSKR
EPI528468_19870/2013	.	.	.	.	D	G	.	.	.	N	.	.	.	.	.	.	KSKR
EPI528702_18224/2012	.	.	.	.	D	G	.	.	.	N	.	.	I	.	.	.	KSKR
EPI462772_11309/2011	.	.	.	.	D	G	.	.	.	N	.	.	.	.	.	.	KSKR
EPI462691_8415/2010	.	.	.	.	D	G	.	.	.	N	.	.	.	.	.	.	KSSR
EPI462631_5209/2009	.	.	.	.	D	G	.	.	.	N	.	.	.	.	.	.	KSSR
EPI383796_FDIL112/2007	.	.	.	.	D	G	.	.	.	N	.	.	.	.	.	.	RSSR
EPI453588_VP01/2006	.	.	.	.	D	G	.	.	.	N	.	.	.	.	.	.	KSSR
EPI985025_G1/1997	.	E	I	R	G	G	F	Y	E	.	V	K	.	S	.	.	RSSR

**Table 2 viruses-15-00461-t002:** Histopathological scoring of different parameters of trachea and lung at experimental infection with LT_2 and LT_56 H9N2 viruses in Sonali and broiler chickens.

Infected Groups	Sonali Chickens						Broiler Chickens				
Lesions/Dpi	1	3	5	7	10	15	1	3	5	7	10
Trachea											
Proliferation of goblet cells	+	+	+	+	+	+	+/−	+	−	++	++
Loss of epithelial layer	+	+	+++	+++	+++	+++	−	+	+++	−	−
Loss of cilia	+	+	+++	+++	+++	+++	−	+	+++	−	−
Inflammation	+/−	+	++	+++	+++	+++	−	+	+++	−	−
Congestion and	−	−	−	−	−	−	+	−	−	−	−
Hemorrhages	−	−	−	−	−	−	−	−	−	−	−
Lung											
Congestion	+	++	++	+++	+	+/−	++	++	+++	+	+
Hemorrhage	+	+	++	+++	+	+/−	++	+	+++	+	+/−
Presence of inflammatory cells	+/−	+/−	+	+	+	+/−	+/−	+/−	++	+	+
Collapsing of alveoli	+	+	+	++	+	+/−	+	++	++	−	−
Rupture of alveoli	−	−	+	++	+	+/−	+	++	++	−	−
Loss of Pneumocyte−I	+++	+++	+++	+++	+	+	+	+	+	−	−
Increased proliferation of pneumocyte−I	−	−	−	−	+	+	−	−	−	−	−
Proliferation of pneumocyte−II	+	++	+++	+	+	+	+	+	+	−	−
Intestine											
Shortening of villi	−	++	−	−	−	−	−	+	−	−	−
Desquamation of villus epithelium	−	−	+++	−	−	−	−	−	++	−	−
Infiltration of inflammatory cells	−	−	++	−	−	−	−	−	+	−	−
Glandular proliferation of crypt epithelium	−	−	++	−	−	−	−	−	+	−	−

## Data Availability

Data will be made available on request.

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
