# Peer review of "Experimental Pathogenicity of H9N2 Avian Influenza Viruses Harboring a Tri-Basic Hemagglutinin Cleavage Site in Sonali and Broiler Chickens"

_viruses, 2023, doi:10.3390/v15020461_

Round 1

Reviewer 1 Report

The manuscript is very well written focusing the experimental pathogenicity of specific H9N2 avian influenza viruses in  Sonali and Broiler chickens. Findings are very convencing and well documented supported by molecular, histopathological, serological ,etc. methods along with self-explanatory figures and suppl. data.

Comments:

In the abstract clearly mention what was the difference if any or no difference observed in the experimental pathogenicity of this  H9N2 avian influenza virus in  Sonali and Broiler chickens. Was there any difference in the pathogenicity of this virus in  Sonali and Broiler, or both were similar susceptible??

Line 160, what is the rationale for using  200 µl of 10^6 EID50/ml of virus for  the infection here?? Is this dose supported by any reference??

Author Response

Point-by-point rebuttal letter

Manuscript ID: viruses-2190373

“Experimental pathogenicity of H9N2 avian influenza viruses harboring a tri-basic hemagglutinin cleavage site in Sonali and Broiler chickens”

Response to reviewer 1

The manuscript is very well written focusing the experimental pathogenicity of specific H9N2 avian influenza viruses in Sonali and Broiler chickens. Findings are very convencing and well documented supported by molecular, histopathological, serological etc. methods along with self-explanatory figures and suppl. data.

So many thanks for the appreciation. We have read the manuscript and addressed the comments to improve.

Comments:

In the abstract clearly mention what was the difference if any or no difference observed in the experimental pathogenicity of this H9N2 avian influenza virus in Sonali and Broiler chickens. Was there any difference in the pathogenicity of this virus in Sonali and Broiler, or both were similar susceptible??

Response: There was no difference in the pathogenicity of these viruses in both groups of birds, which has been added now in the line 28-30 (track change) in the abstract section.

Line 160, what is the rationale for using 200 µl of 10^6 EID50/ml of virus for the infection here? Is this dose supported by any reference??

Response:  In several studies 200 µl of 10^6 EID50/ml of virus was used for the infection including Munir et al., 2013 or Arnold et al., 2013. These references have now been added in the text in line 164 (Ref. 26, 27).

 ID: viruses-2190373

“Experimental pathogenicity of H9N2 avian influenza viruses harboring a tri-basic hemagglutinin cleavage site in Sonali and Broiler chickens”

Response to reviewer 1

The manuscript is very well written focusing the experimental pathogenicity of specific H9N2 avian influenza viruses in Sonali and Broiler chickens. Findings are very convencing and well documented supported by molecular, histopathological, serological etc. methods along with self-explanatory figures and suppl. data.

So many thanks for the appreciation. We have read the manuscript and addressed the comments to improve.

Comments:

In the abstract clearly mention what was the difference if any or no difference observed in the experimental pathogenicity of this H9N2 avian influenza virus in Sonali and Broiler chickens. Was there any difference in the pathogenicity of this virus in Sonali and Broiler, or both were similar susceptible??

Response: There was no difference in the pathogenicity of these viruses in both groups of birds, which has been added now in the line 28-30 (track change) in the abstract section.

Line 160, what is the rationale for using 200 µl of 10^6 EID50/ml of virus for the infection here? Is this dose supported by any reference??

Response:  In several studies 200 µl of 10^6 EID50/ml of virus was used for the infection including Munir et al., 2013 or Arnold et al., 2013. These references have now been added in the text in line 164 (Ref. 26, 27).

Reviewer 2 Report

This paper describes the pathogenicity of the H9N2 low pathogenic avian influenza viruses in chickens raised in Bangladesh. The novelty of this paper is the higher virulence of the H9N2 viruses for the chickens of the strain Sonali, although in terms of the pathogenicity assessment, several similar studies using the H9N2 virus have already been reported. To improve the quality, this reviewer suggests major modifications and clarification of the methodology for this manuscript.

1. Fig. S3: The chickens used in this experiment are described in the materials and methods as seronegative against avian influenza virus. However, this figure shows that 2HI-4HI antibodies in several chickens. This means that these chickens were not seronegative.

2. No detail genetic information for the broiler chicken.

3. Table 2 and Fig. 6: No statistical validation to discuss the pathogenicity of H9N2 viruses for Sonali and broilers was performed.

4. Fig. 6: Why the Ct value of 1 dpi with the real-time RT-PCR for the cloaca swab was zero?

5. Fig. 1: Are the picture of embryonic chicken egg upside down? Additionally, the c in chicken in the virus strain name should be in the lower case.

6. Fig. 2: The HA gene of the H9N2 virus can be roughly classified into Eurasian and North American lineages. The Eurasian lineage is then divided into G1, Y280, and Y439 sublineages. These classifications are not shown in the phylogenetic tree for the HA gene. Additionally, similar genetic classifications should be shown for the NA and other internal genes.

7. Fig. S2: The hemorrhage in the trachea is not visible in the photograph currently shown. It should be changed to a more magnified photo.

8. Several grammatical errors are found in the text.

81 L. 88: RT-PCR

82 L. 93: 80, not -80

83 Fig. 1 and others: 50 is the subscript for "EID50"

Author Response

Point-by-point rebuttal letter

Manuscript ID: viruses-2190373

“Experimental pathogenicity of H9N2 avian influenza viruses harboring a tri-basic hemagglutinin cleavage site in Sonali and Broiler chickens”

Response to reviewer 2

This paper describes the pathogenicity of the H9N2 low pathogenic avian influenza viruses in chickens raised in Bangladesh. The novelty of this paper is the higher virulence of the H9N2 viruses for the chickens of the strain Sonali, although in terms of the pathogenicity assessment, several similar studies using the H9N2 virus have already been reported. To improve the quality, this reviewer suggests major modifications and clarification of the methodology for this manuscript.

So many thanks for the appreciation and critical review of the manuscript. We have amended the manuscript and addressed the comments to improve.

  1. S3: The chickens used in this experiment are described in the materials and methods as seronegative against avian influenza virus. However, this figure shows that 2HI-4HI antibodies in several chickens. This means that these chickens were not seronegative.

Response: The reviewer observation’s is very much appreciated. SPF chicks are not available in Bangladesh. Therefore, we purchased commercial day-old chicks having a high level of maternal derived antibody (MDA) and raised them in isolation with strict biosecurity measures. No AIV infections occurred during the raising period as evidenced by repeated negative swab samples. Before challenge at week 3 and 4 for broiler and Sonali respectively, we measured the level of antibodies in birds. Most of the birds were found seronegative. Few chicks, however, still showed detectable titers below the level that was considered to be protective (less than 5 HI unit). These antibodies are considered to be of passive (MDA) origin. A detailed description is now available in line 390—400. Post challenge titers did increase significantly as shown in Figure S3.

  1. No detail genetic information for the broiler chicken

Response: Cob 500 broilers were used for the experiment (added in line 86).

  1. Table 2 and Fig. 6: No statistical validation to discuss the pathogenicity of H9N2 viruses for Sonali and broilers was performed.

Response: Although we have several animals on our experiment, but we could not proceed with too many samples for histopathology due to financial obligation to see the statistical significance. However, we did statistical analysis for viral load calculation (Fig.6) which is included in line 372-376 (yellow shade and track change).

  1. 6: Why the Ct value of 1 dpi with the real-time RT-PCR for the cloaca swab was zero?

Response: The lack of viral RNA found in cloacal swabs taken at 1 dpi indicates that there was no cloacal shedding on that particular dpi, and this is further supported by the histopathogical findings of the intestine, which show almost normal intestinal morphology and now is included in line 468-470 (yellow shade and track change)

  1. 1: Are the picture of embryonic chicken egg upside down? Additionally, the “c” in chicken in the virus strain name should be in the lower case.

Response: We have revised the figure accordingly (Fig 1; 172).

  1. 2: The HA gene of the H9N2 virus can be roughly classified into Eurasian and North American lineages. The Eurasian lineage is then divided into G1, Y280, and Y439 sublineages. These classifications are not shown in the phylogenetic tree for the HA gene. Additionally, similar genetic classifications should be shown for the NA and other internal genes.

Response: The Eurasian lineage is divided into three further classes, those are already included on the phylogenetic tree of HA, NA and other internal genes (Fig. 2 and Suppl. Fig 1). We just highlighted the G1 strain as all of the four studied isolates belong to this sub-lineage. Therefore, the other sub-lineages might not be necessarily highlighted here.

  1. S2: The hemorrhage in the trachea is not visible in the photograph currently shown. It should be changed to a more magnified photo.

Response:   As we mentioned “slight hemorrhages along the tracheal ring shown up to 15 dpi” (in line 282-284) can be seen in a closer view in figure S2. The image quality decrease with increased magnification.

  1. Several grammatical errors are found in the text.

8–1 L. 88: RT-PCR

8–2 L. 93: –80, not -80

8–3 Fig. 1 and others: 50 is the subscript for "EID50"

Response: We have revised the grammatical errors accordingly (yellow shade and track change)

Round 2

Reviewer 2 Report

1.     Table 2 and Fig. 6: No statistical validation to discuss the pathogenicity of H9N2 viruses for Sonali and broilers was performed.

Response: Although we have several animals on our experiment, but we could not proceed with too many samples for histopathology due to financial obligation to see the statistical significance. However, we did statistical analysis for viral load calculation (Fig.6) which is included in line 372-376 (yellow shade and track change).

Comment from this reviewer: I understand that you did statistical analysis for Fig. 6; however, you did not indicate specific points of your significance in Fig. 6 (e.g. astarisk).

2. Why the Ct value of 1 dpi with the real-time RT-PCR for the cloaca swab was zero?

 Response: The lack of viral RNA found in cloacal swabs taken at 1 dpi indicates that there was no cloacal shedding on that particular dpi, and this is further supported by the histopathogical findings of the intestine, which show almost normal intestinal morphology and now is included in line 468-470 (yellow shade and track change)

 Comment from this reviewer: If you did not detect any viral RNA by RT-qPCR, the result should be “more than 40” Ct value (>40), but not “zero”.

3.     The HA gene of the H9N2 virus can be roughly classified into Eurasian and North American lineages. The Eurasian lineage is then divided into G1, Y280, and Y439 sublineages. These classifications are not shown in the phylogenetic tree for the HA gene. Additionally, similar genetic classifications should be shown for the NA and other internal genes.

 Response: The Eurasian lineage is divided into three further classes, those are already included on the phylogenetic tree of HA, NA and other internal genes (Fig. 2 and Suppl. Fig 1). We just highlighted the G1 strain as all of the four studied isolates belong to this sub-lineage. Therefore, the other sub-lineages might not be necessarily highlighted here.

 Comment from this reviewer: The nomenclature of the representative gene lineages should be shown first to give the reader an overall picture of the phylogenetic tree. Then, information should be added so that the clusters of Bangladeshi virus strains in the G1 lineage, which is the focus of this paper, can be easily understood.

*Figure 1 in the following paper will help you do that.

https://doi.org/10.4142/jvs.2021.22.e21

4. The English grammar, especially around the sentences you have just corrected, needs to be thoroughly revised. In addition, the format of your reference part also needs to be modified, as it is not at all well formatted.

Author Response

Revision2 Manuscript ID: viruses-2190373

1. Table 2 and Fig. 6: No statistical validation to discuss the pathogenicity of H9N2 viruses for Sonali and broilers was performed.

Response: Although we have several animals on our experiment, but we could not proceed with too many samples for histopathology due to financial obligation to see the statistical significance. However, we did statistical analysis for viral load calculation (Fig.6) which is included in line 372-376 (yellow shade and track change).

Comment from this reviewer: I understand that you did statistical analysis for Fig. 6; however, you did not indicate specific points of your significance in Fig. 6 (e.g. astarisk).

Response 2: Thanks for your understanding. We now have spotted the asterisk mark and added the information in figure legend.

2. Why the Ct value of 1 dpi with the real-time RT-PCR for the cloaca swab was zero?

Response: The lack of viral RNA found in cloacal swabs taken at 1 dpi indicates that there was no cloacal shedding on that particular dpi, and this is further supported by the histopathological findings of the intestine, which show almost normal intestinal morphology and now is included in line 468-470 (yellow shade and track change)

Comment from this reviewer: If you did not detect any viral RNA by RT-qPCR, the result should be “more than 40” Ct value (>40), but not “zero”.

Response 2: The samples that showed value of > 40 considered as negative and in the graph represent as zero, now added in line 381-382

3. The HA gene of the H9N2 virus can be roughly classified into Eurasian and North American lineages. The Eurasian lineage is then divided into G1, Y280, and Y439 sublineages. These classifications are not shown in the phylogenetic tree for the HA gene. Additionally, similar genetic classifications should be shown for the NA and other internal genes.

Response: The Eurasian lineage is divided into three further classes, those are already included on the phylogenetic tree of HA, NA and other internal genes (Fig. 2 and Suppl. Fig 1). We just highlighted the G1 strain as all of the four studied isolates belong to this sub-lineage. Therefore, the other sub-lineages might not be necessarily highlighted here.

Comment from this reviewer: The nomenclature of the representative gene lineages should be shown first to give the reader an overall picture of the phylogenetic tree. Then, information should be added so that the clusters of Bangladeshi virus strains in the G1 lineage, which is the focus of this paper, can be easily understood. *Figure 1 in the following paper will help you do that. https://doi.org/10.4142/jvs.2021.22.e21

Response 2: Thank you for your suggestion, classification has now shown in Figure 2 HA phylogeny.

4. The English grammar, especially around the sentences you have just corrected, needs to be thoroughly revised. In addition, the format of your reference part also needs to be modified, as it is not at all well formatted.

Response 2: So sorry for not looking at the references. Now we have worked on all the references (yellow shade). The manuscript has been overlooked by a native English speaker in the USA (where I am currently staying).